# Learning Retrospective Knowledge with Reverse Reinforcement Learning

**Shangtong Zhang** *
University of Oxford

**Vivek Veeriah**
University of Michigan, Ann Arbor

**Shimon Whiteson**
University of Oxford

## Abstract

We present a Reverse Reinforcement Learning (Reverse RL) approach for representing *retrospective knowledge*. General Value Functions (GVFs) have enjoyed great success in representing *predictive knowledge*, i.e., answering questions about possible future outcomes such as "how much fuel will be consumed in expectation if we drive from A to B?". GVFs, however, cannot answer questions like "how much fuel do we expect a car to have given it is at B at time $t$?". To answer this question, we need to know when that car had a full tank and how that car came to B. Since such questions emphasize the influence of possible past events on the present, we refer to their answers as *retrospective knowledge*. In this paper, we show how to represent retrospective knowledge with Reverse GVFs, which are trained via Reverse RL. We demonstrate empirically the utility of Reverse GVFs in both representation learning and anomaly detection.

## 1 Introduction

Much knowledge can be formulated as answers to predictive questions (Sutton, 2009), for example, "to know that Joe is in the coffee room is to predict that you will see him if you went there" (Sutton, 2009). Such knowledge is referred to as *predictive knowledge* (Sutton, 2009; Sutton et al., 2011). General Value Functions (GVFs, Sutton et al. 2011) are commonly used to represent predictive knowledge. GVFs are essentially the same as canonical value functions (Puterman, 2014; Sutton and Barto, 2018).

However, the policy, the reward function, and the discount function associated with GVFs are usually carefully designed such that the numerical value of a GVF at certain states matches the numerical answer to certain predictive questions. In this way, GVFs can represent predictive knowledge.

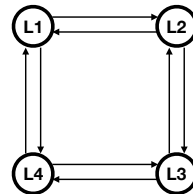

Consider the concrete example in Figure 1, where a microdrone is doing a random walk. The microdrone is initialized somewhere with 100% battery. L4 is a power station where its battery is recharged to 100%. Each clockwise movement consumes 2% of the battery, and each counterclockwise movement consumes 1% (for simplicity, we assume negative battery levels, e.g., -10%, are legal). Furthermore, each movement fails with probability 1%, in which case the microdrone remains in the same location and no energy is consumed. An example of a predictive question in this system is:

Figure 1: A microdrone doing random walk among four different locations. L4 is a charging station where the microdrone's battery is fully recharged.

**Question 1.** *Starting from L1, how much energy will be consumed in expectation before the next charge?*

To answer this question, we can model the system as a Markov Decision Process (MDP). The policy is uniformly random and the reward for each movement is the

additive inverse of the corresponding battery consumption. Whenever the microdrone reaches state L4, the episode terminates. Under this setup, the answer to Question 1 is the expected cumulative reward when starting from L1, i.e., the state value of L1. Hence, GVFs can represent the predictive knowledge in Question 1. As a GVF is essentially a value function, it can be trained with any data stream from agent-environment interaction via Reinforcement Learning (RL, Sutton and Barto 2018), demonstrating the generality of the GVF approach. Importantly, the most appealing feature of GVFs is their compatibility with off-policy learning, making this representation of predictive knowledge scalable and efficient. For example, in the Horde architecture (Sutton et al., 2011), many GVFs are learned in parallel with gradient-based off-policy temporal difference methods (Sutton et al., 2009b,a; Maei, 2011). In the microdrone example, we can learn the answer to Question 1 under many different conditions (e.g., when the charging station is located at L2 or when the microdrone moves clockwise with probability 80%) simultaneously with off-policy learning by considering different reward functions, discount functions, and polices.

GVFs, however, cannot answer many other useful questions, e.g., if at some time $t$, we find the microdrone at L1, how much battery do we expect it to have? As such questions emphasize the influence of possible past events on the present, we refer to their answers as *retrospective knowledge*. Such retrospective knowledge is useful, for example, in anomaly detection. Suppose the microdrone runs for several weeks by itself while we are traveling. When we return at time $t$, we find the microdrone is at L1. We can then examine the battery level and see if it is similar to the expected battery at L1. If there is a large difference, it is likely that there is something wrong with the microdrone. There are, of course, many methods to perform such anomaly detection. For example, we could store the full running log of the microdrone during our travel and examine it when we are back. The memory requirement to store the full log, however, increases according to the length of our travel. By contrast, if we have retrospective knowledge, i.e., the expected battery level at each location, we can program the microdrone to log its battery level at each step (overwriting the record from the previous step). We can then examine the battery level when we are back and see if it matches our expectation. The current battery level can be easily computed via the previous battery level and the energy consumed at the last step, using only constant computation per step. The storage of the battery level requires only constant memory as we do not need to store the full history, which would not be feasible for a microdrone. Thus retrospective knowledge provides a memory-efficient way to perform anomaly detection. Of course, this approach may have lower accuracy than storing the full running log. This is indeed a trade-off between accuracy and memory, and we expect applications of this approach in memory-constrained scenarios such as embedded systems.

To know the expected battery level at L1 at time $t$ is essentially to answer the following question:

**Question 2.** *How much energy do we expect the microdrone to have consumed since the last time it had 100% battery given that it is at L1 at time $t$?*

Unfortunately, GVFs cannot represent retrospective knowledge (e.g., the answer to Question 2) easily. GVFs provide a mechanism to ignore all future events after reaching certain states via setting the discount function at those states to be 0. This mechanism is useful for representing predictive knowledge. For example, in Question 1, we do not care about events *after* the next charge. For retrospective knowledge, we, however, need a mechanism to ignore all previous events before reaching certain states. For example, in Question 2, we do not care about events *before* the last time the microdrone had 100% battery. Unfortunately, GVFs do not have such a mechanism. In Appendix A, we describe several tricks that attempt to represent retrospective knowledge with GVFs and explain why they are invalid.

In this paper, we propose *Reverse GVFs* to represent retrospective knowledge. Using the same MDP formulation of the microdrone system, let the random variable $\bar{G}_t$ denote the energy the microdrone has consumed at time $t$ since the last time it had 100% battery. To answer Question 2, we are interested in the conditional expectation of $\bar{G}_t$ given that $S_t = $ L1. We refer to functions describing such conditional expectations as Reverse GVFs, which we propose to learn via *Reverse Reinforcement Learning*. The key idea of Reverse RL is still bootstrapping, but in the reverse direction. It is easy to see that $\bar{G}_t$ depends on $\bar{G}_{t-1}$ and the energy consumption from $t-1$ to $t$. In general, the quantity of interest at time $t$ depends on that at time $t-1$ in Reverse RL. This idea of bootstrapping from the past has been explored by Wang et al. (2007, 2008); Hallak and Mannor (2017); Gelada and Bellemare (2019); Zhang et al. (2020d) but was limited to the density ratio learning setting. We propose several Reverse RL algorithms and prove their convergence under linear function approximation. We also propose Distributional Reverse RL algorithms akin to Distributional

RL (Bellemare et al., 2017; Dabney et al., 2017; Rowland et al., 2018) to compute the probability of an event for anomaly detection. We demonstrate empirically the utility of Reverse GVFs in anomaly detection and representation learning.

Besides Reverse RL, there are other approaches we could consider for answering Question 2. For example, we could formalize it as a simple regression task, where the input is the location and the target is the power consumption since the last time the microdrone had 100% battery. We show below that this regression formulation is a special case of Reverse RL, similar to how Monte Carlo is a special case of temporal difference learning (Sutton, 1988). Alternatively, answering Question 2 is trivial if we have formulated the system as a Partially Observable MDP. We could use either the location or the battery level as the state and the other as the observation. In either case, however, deriving the conditional observation probabilities is nontrivial. We could also model the system as a reversed chain directly as Morimura et al. (2010) in light of reverse bootstrapping. This, however, creates difficulties in off-policy learning, which we discuss in Section 5.

## 2    Background

We consider an infinite-horizon Markov Decision Process (MDP) with a finite state space $\mathcal{S}$, a finite action space $\mathcal{A}$, a transition kernel $p : \mathcal{S} \times \mathcal{S} \times \mathcal{A} \to [0, 1]$, and an initial distribution $\mu_0 : \mathcal{S} \to [0, 1]$. In the GVF framework, users define a reward function $r : \mathcal{S} \times \mathcal{A} \to \mathbb{R}$, a discount function $\gamma : \mathcal{S} \to [0, 1]$, and a policy $\pi : \mathcal{A} \times \mathcal{S} \to [0, 1]$ to represent certain predictive questions. An agent is initialized at $S_0$ according to $\mu_0$. At time step $t$, an agent at a state $S_t$ selects an action $A_t$ according to $\pi(\cdot|S_t)$, receives a bounded reward $R_{t+1}$ satisfying $\mathbb{E}[R_{t+1}] = r(S_t, A_t)$, and proceeds to the next state $S_{t+1}$ according to $p(\cdot|S_t, A_t)$. We then define the return at time step $t$ recursively as

$$G_t \doteq R_{t+1} + \gamma(S_{t+1})G_{t+1},$$

which allows us to define the general value function $v_\pi(s) \doteq \mathbb{E}[G_t|S_t = s]$.[2] The general value function $v_\pi$ is essentially the same as the canonical value function (Puterman, 2014; Sutton and Barto, 2018). The name "general" emphasizes its usage in representing predictive knowledge. In the microdrone example (Figure 1), we define the reward function as $r(s, a_1) = 2, r(s, a_2) = 1 \ \forall s$, where $a_1$ is moving clockwise and $a_2$ is moving counterclockwise. We define the discount function as $\gamma(\text{L1}) = \gamma(\text{L2}) = \gamma(\text{L3}) = 1, \gamma(\text{L4}) = 0$. Then it is easy to see that the numerical value of $v_\pi(\text{L1})$ is the answer to Question 1. In the rest of the paper, we use functions and vectors interchangeably, e.g., we also interpret $v_\pi$ as a vector in $\mathbb{R}^{|\mathcal{S}|}$. Furthermore, all vectors are column vectors.

The general value function $v_\pi$ is the unique fixed point of the generalized Bellman operator $\mathcal{T}$ (Yu et al., 2018): $\mathcal{T}y \doteq r_\pi + P_\pi \Gamma y$, where $P_\pi \in \mathbb{R}^{|\mathcal{S}| \times |\mathcal{S}|}$ is the state transition matrix, i.e., $P_\pi(s, s') \doteq \sum_a \pi(a|s)p(s'|s, a)$, $r_\pi \in \mathbb{R}^{|\mathcal{S}|}$ is the reward vector, i.e., $r_\pi(s) \doteq \sum_a \pi(a|s)r(s, a)$, and $\Gamma \in \mathbb{R}^{|\mathcal{S}| \times |\mathcal{S}|}$ is a diagonal matrix whose $s$-th diagonal entry is $\gamma(s)$. To ensure $v_\pi$ is well-defined, we assume $\pi$ and $\gamma$ are defined such that $(I - P_\pi \Gamma)^{-1}$ exists (Yu, 2015). Then if we interpret $1 - \gamma(s)$ as the probability for an episode to terminate at $s$, we can assume termination occurs w.p. 1.

## 3    Reverse General Value Function

Inspired by the return $G_t$, we define the reverse return $\bar{G}_t$, which accumulates previous rewards:

$$\bar{G}_t \doteq R_t + \gamma(S_{t-1})\bar{G}_{t-1}, \quad \bar{G}_0 \doteq 0.$$

In the reverse return $\bar{G}_t$, the discount function $\gamma$ has different semantics than in the return $G_t$. Namely, in $G_t$, the discount function down-weights future rewards, while in $\bar{G}_t$, the discount function down-weights past rewards. In an extreme case, setting $\gamma(S_{t-1}) = 0$ allows us to ignore all the rewards before time $t$ when computing the reverse return $\bar{G}_t$, which is exactly the mechanism we need to represent retrospective knowledge.

Let us consider the microdrone example again (Figure 1) and try to answer Question 2. Assume the microdrone was initialized at L3 at $t = 0$ and visited L4 and L1 afterwards. Then it is easy to

see that $\bar{G}_2$ is exactly the energy the microdrone has consumed since its last charge. In general, if we find the microdrone at L1 at time $t$, the expectation of the energy that the microdrone has consumed since its last charge is exactly $\mathbb{E}_{\pi,p,r}[\bar{G}_t|S_t = \text{L1}]$. Note the answer to Question 2 is not homogeneous in $t$. For example, suppose the microdrone is initialized at L4 at $t = 0$. If we find it at L1 at $t = 1$, it is trivial to see the microdrone has consumed 2% battery. By contrast, if we find it at L1 at $t = 100$, computing the energy consumption since the last time it had 100% battery is nontrivial. It is inconvenient that the answer depends the time step $t$ but fortunately, we can show the following:

**Assumption 1.** *The chain induced by $\pi$ is ergodic and $(I - P_\pi^\top \Gamma)^{-1}$ exists.*

**Theorem 1.** *Under Assumption 1, the limit $\lim_{t\to\infty} \mathbb{E}[\bar{G}_t|S_t = s]$ exists, which we refer to as $\bar{v}_\pi(s)$. Furthermore, we define the reverse Bellman operator $\bar{\mathcal{T}}$ as*

$$\bar{\mathcal{T}}y \doteq D_\pi^{-1}\tilde{P}_\pi^\top \tilde{D}_\pi r + D_\pi^{-1} P_\pi^\top \Gamma D_\pi y,$$

*where $D_\pi \doteq diag(d_\pi) \in \mathbb{R}^{|\mathcal{S}|\times|\mathcal{S}|}$ with $d_\pi$ being the stationary distribution of the chain induced by $\pi$, $\tilde{P}_\pi \in \mathbb{R}^{|\mathcal{S}||\mathcal{A}|\times|\mathcal{S}|}$ is the transition matrix, i.e., $\tilde{P}_\pi((s,a), s') \doteq p(s'|s,a)$, and $\tilde{D}_\pi \doteq diag(\tilde{d}_\pi) \in \mathbb{R}^{|\mathcal{S}||\mathcal{A}|\times|\mathcal{S}||\mathcal{A}|}$ with $\tilde{d}_\pi(s,a) \doteq d_\pi(s)\pi(a|s)$. Then $\bar{\mathcal{T}}$ is a contraction mapping w.r.t. some weighted maximum norm, and $\bar{v}_\pi$ is its unique fixed point. We have $\bar{v}_\pi = D_\pi^{-1}(I - P_\pi^\top \Gamma)^{-1}\tilde{P}_\pi^\top \tilde{D}_\pi r$.*

Assumption 1 can be easily fulfilled in the real world as long as the problem we consider has a recurring structure. The proof of Theorem 1 is based on Sutton et al. (2016); Zhang et al. (2019, 2020d) and is detailed in the appendix. Theorem 1 states that the numerical value of $\bar{v}_\pi(\text{L1})$ approximately answers Question 2. When Question 2 is asked for a large enough $t$, the error in the answer $\bar{v}_\pi(\text{L1})$ is arbitrarily small. We call $\bar{v}_\pi(s)$ a *Reverse General Value Function*, which approximately encodes the retrospective knowledge, i.e., the answer to the retrospective question induced by $\pi, r, \gamma, t$ and $s$.

Based on the reverse Bellman operator $\bar{\mathcal{T}}$, we now present the Reverse TD algorithm. Let us consider linear function approximation with a feature function $x : \mathcal{S} \to \mathbb{R}^K$, which maps a state to a $K$-dimensional feature. We use $X \in \mathbb{R}^{|\mathcal{S}|\times K}$ to denote the feature matrix, each row of which is $x(s)^\top$. Our estimate for $\bar{v}_\pi$ is then $Xw$, where $w \in \mathbb{R}^K$ contains the learnable parameters. At time step $t$, Reverse TD computes $w_{t+1}$ as

$$w_{t+1} \doteq w_t + \alpha_t(R_t + \gamma(S_{t-1})x_{t-1}^\top w_t - x_t^\top w_t)x_t, \tag{1}$$

where $x_t \doteq x(S_t)$ is shorthand, and $\{\alpha_t\}$ is a deterministic positive nonincreasing sequence satisfying the Robbins-Monro condition (Robbins and Monro, 1951), i.e., $\sum_t \alpha_t = \infty, \sum_t^\infty \alpha_t^2 < \infty$. We have

**Proposition 1.** *(Convergence of Reverse TD) Under Assumption 1, assuming $X$ has linearly independent columns, then the iterate $\{w_t\}$ generated by Reverse TD (Eq (1)) satisfies $\lim_{t\to\infty} w_t = -\bar{A}^{-1}\bar{b}$ with probability 1, where $\bar{A} \doteq X^\top(P_\pi^\top \Gamma - I)D_\pi X, \bar{b} \doteq X^\top \tilde{P}_\pi^\top \tilde{D}_\pi r$.*

The proof of Proposition 1 is based on the proof of the convergence of linear TD in Bertsekas and Tsitsiklis (1996). In particular, we need to show that $\bar{A}$ is negative definite. Details are provided in the appendix. For a sanity check, it is easy to verify that in the tabular setting (i.e., $X = I$), $-\bar{A}^{-1}\bar{b} = \bar{v}_\pi$ indeed holds. Inspired by the success of TD($\lambda$) (Sutton, 1988) and COP-TD($\lambda$) (Hallak and Mannor, 2017), we also extend Reverse TD to Reverse TD($\lambda$), which updates $w_{t+1}$ as

$$w_{t+1} \doteq w_t + \alpha_t\Big(R_t + \gamma(S_{t-1})\big((1-\lambda)x_{t-1}^\top w_t + \lambda\bar{G}_{t-1}\big) - x_t^\top w_t\Big)x_t.$$

With $\lambda = 1$, Reverse TD($\lambda$) reduces to supervised learning.

**Distributional Learning.** In anomaly detection with Reverse GVFs, we compare the observed quantity (a scalar) with our retrospective knowledge (a scalar, the conditional expectation). It is not clear how to translate the difference between the two scalars into a decision about whether there is an anomaly. If our retrospective knowledge is a distribution instead, we can perform anomaly detection from a probabilistic perspective. To this end, we propose Distributional Reverse TD, akin to Bellemare et al. (2017); Rowland et al. (2018).

We use $\eta_t^s \in \mathcal{P}(\mathbb{R})$ to denote the conditional probability distribution of $\bar{G}_t$ given $S_t = s$, where $\mathcal{P}(\mathbb{R})$ is the set of all probability measures over the measurable space $(\mathbb{R}, \mathcal{B}(\mathbb{R}))$, with $\mathcal{B}(\mathbb{R})$ being the Borel sets of $\mathbb{R}$. Moreover, we use $\eta_t \in (\mathcal{P}(\mathbb{R}))^{|\mathcal{S}|}$ to denote the vector whose $s$-th element is $\eta_t^s$. By the definition of $\bar{G}_t$, we have for any $E \in \mathcal{B}(\mathbb{R})$

$$\eta_t^s(E) = \int_{\mathbb{R}\times\mathcal{S}}(f_{r,\bar{s}}\#\eta_{t-1}^{\bar{s}})(E)\mathrm{d}\Pr(S_{t-1} = \bar{s}, R_t = r|S_t = s), \tag{2}$$

where $f_{r,\bar{s}} : \mathbb{R} \to \mathbb{R}$ is defined as $f_{r,\bar{s}}(x) = r + \gamma(\bar{s})x$, and $f_{r,\bar{s}}\#\eta_{t-1}^{\bar{s}} : \mathcal{B}(\mathbb{R}) \to [0,1]$ is the push-forward measure, i.e., $(f_{r,\bar{s}}\#\eta_{t-1}^{\bar{s}})(E) \doteq \eta_{t-1}^{\bar{s}}(f_{r,\bar{s}}^{-1}(E))$, where $f_{r,\bar{s}}^{-1}(E)$ is the preimage of $E$. To study $\eta_t^s$ when $t \to \infty$, we define

$$p(\bar{s}, r|s) \doteq \lim_{t\to\infty} \Pr(S_{t-1} = \bar{s}, R_t = r|S_t = s) = \frac{d_\pi(\bar{s})}{d_\pi(s)} \sum_{\bar{a}} \pi(\bar{a}|\bar{s})p(s|\bar{s}, \bar{a}) \Pr(r|\bar{s}, \bar{a}).$$

When $t \to \infty$, Eq (2) suggests $\eta_t^s(E)$ evolves according to $\eta_t^s(E) = \int_{\mathbb{R}\times\mathcal{S}} (f_{r,\bar{s}}\#\eta_{t-1}^{\bar{s}})(E)\mathrm{d}\,p(\bar{s}, r|s)$. We, therefore, define the distributional reverse Bellman operator $\tilde{\mathcal{T}} : (\mathcal{P}(\mathbb{R}))^{|\mathcal{S}|} \to (\mathcal{P}(\mathbb{R}))^{|\mathcal{S}|}$ as $(\tilde{\mathcal{T}}\eta)^s \doteq \int_{\mathbb{R}\times\mathcal{S}} (f_{r,\bar{s}}\#\eta^{\bar{s}})\mathrm{d}\,p(\bar{s}, r|s)$. We have

**Proposition 2.** *Under Assumption 1, $\tilde{\mathcal{T}}$ is a contraction mapping w.r.t. a metric $d$, and we refer to its fixed point as $\eta_\pi$. Assuming $\mu_0 = d_\pi$, then $\lim_{t\to\infty} d(\eta_t, \eta_\pi) = 0$.*

We now provide a practical algorithm to approximate $\eta_\pi^s$ based on quantile regression, akin to Dabney et al. (2017). We use $N$ quantiles with quantile levels $\{\tau_i\}_{i=1,\dots,N}$, where $\tau_i \doteq \frac{(i-1)/N + i/N}{2}$. The measure $\eta_\pi^s$ is approximated with $\frac{1}{N} \sum_{i=1}^N \delta_{q_i(s;\theta)}$, where $\delta_x$ is a Dirac at $x$, $q_i(s;\theta)$ is a quantile corresponding to the quantile level $\tau_i$, and $\theta$ is learnable parameters. Given a transition $(s, a, r, s')$, we train $\theta$ to minimize the following quantile regression loss

$$L(\theta) \doteq \sum_{i=1}^N \sum_{j=1}^N \rho_{\tau_i}^\kappa \left( r + \frac{\gamma(s)}{N} \sum_{k=1}^N q_j(s;\bar{\theta}) - \frac{1}{N} \sum_{k=1}^N q_i(s';\theta) \right),$$

where $\bar{\theta}$ contains the parameters of the target network (Mnih et al., 2015), which is synchronized with $\theta$ periodically, and $\rho_{\tau_i}^\kappa(x) \doteq |\tau_i - \mathbb{I}_{x<0}|\mathcal{H}_\kappa(x)$ is the quantile regression loss function. $\mathcal{H}_\kappa(x)$ is the Huber loss, i.e., $\mathcal{H}_\kappa(x) \doteq 0.5x^2\mathbb{I}_{x\leq\kappa} + \kappa(|x| - 0.5\kappa)\mathbb{I}_{x>\kappa}$, where $\kappa$ is a hyperparameter. Dabney et al. (2017) provide more details about quantile-regression-based distributional RL.

**Off-policy Learning.** We would also like to be able to answer to Question 2 without making the microdrone do a random walk, i.e., we may have another policy $\mu$ for the microdrone to collect data. In this scenario, we want to learn $\bar{v}_\pi$ off-policy. We consider Off-policy Reverse TD, which updates $w_t$ as:

$$w_{t+1} \doteq w_t + \alpha_t \tau(S_{t-1})\rho(S_{t-1}, A_{t-1})(R_t + \gamma(S_{t-1})x_{t-1}^\top w_t - x_t^\top w_t)x_t, \tag{3}$$

where $\tau(s) \doteq \frac{d_\pi(s)}{d_\mu(s)}, \rho(s, a) \doteq \frac{\pi(a|s)}{\mu(a|s)}$ and $\{S_0, A_0, R_1, S_1, \dots\}$ is obtained by following the behavior policy $\mu$. Here we assume access to the density ratio $\tau(s)$, which can be learned via Hallak and Mannor (2017); Gelada and Bellemare (2019); Nachum et al. (2019); Zhang et al. (2020a,c).

**Proposition 3.** *(Convergence of Off-policy Reverse TD) Under Assumption 1, assuming $X$ has linearly independent columns, and the chain induced by $\mu$ is ergodic, then the iterate $\{w_t\}$ generated by Off-policy Reverse TD (Eq (3)) satisfies $\lim_{t\to\infty} w_t = -\bar{A}^{-1}\bar{b}$ with probability 1.*

Off-policy Reversed TD converges to the same point as on-policy Reverse TD. This convergence relies heavily on having the true density ratio $\tau(s)$. When using a learned estimate for the density ratio, approximation error is inevitable and thus convergence is not ensured. It is straightforward to consider a GTD (Sutton et al., 2009b,a; Maei, 2011) analogue, Reverse GTD, similar to Gradient Emphasis Learning in Zhang et al. (2020d). The convergence of Off-Policy Reverse GTD is straightforward (Zhang et al., 2020d), but to a different point from On-policy Reverse TD.

## 4 Experiments

**The Effect of $\lambda$.** [3] At time step $t$, the reverse return $\bar{G}_t$ is known and can approximately serve as a sample for $\bar{v}_\pi(S_t)$. It is natural to model this as a regression task where the input is $S_t$, and the target is $\bar{G}_t$. This is indeed Reverse TD(1). So we first study the effect of $\lambda$ in Reverse TD($\lambda$). We consider the microdrone example in Figure 1. The dynamics are specified in Section 1. The reward function and the discount function are specified in Section 2. The policy $\pi$ is uniformly random. We use a tabular representation and compute the ground truth $\bar{v}_\pi$ analytically. We vary $\lambda$ in $\{0, 0.3, 0.7, 0.9, 1.0\}$. For each $\lambda$, we use a constant step size $\alpha$ tuned from $\{10^{-3}, 5 \times 10^{-3}, 10^{-2}, 5 \times 10^{-2}\}$. We report the Mean Value Error (MVE) against training steps in Figure 2. At a time step $t$, assuming our estimation is $\bar{V}$, the MVE is computed as $||\bar{V} - \bar{v}_\pi||_2^2$. The results show that the bias of the estimate decreases quickly at the beginning. As a result, variance of the update target becomes the major obstacle in the learning process, which explains why the best performance is achieved by smaller $\lambda$ in this experiment.

**Anomaly Detection.** [3]

*Tabular Representation.* Consider the microdrone example once again (Figure 1). Suppose we want the microdrone to follow a policy $\pi$ where $\pi(a_1|s) = 0.1 \, \forall s$. However, something can go wrong when the microdrone is following $\pi$. For example, it may start to take $a_1$ with probability 0.9 at all states due to a malfunctioning navigation system, which we refer to as a *policy anomaly*. The microdrone may also consume 2% extra battery per step with probability 0.5 due to a malfunctioning engine, which we refer to as a *reward anomaly*, i.e., the reward $R_t$ becomes $R_t + 2$ with probability 0.5. We cannot afford to monitor the microdrone every time step but can do so occasionally, and we hope if something has gone wrong we can discover it. Since it is a microdrone, it does not have the memory to store all the logs between examinations. We now demonstrate that Reverse GVFs can discover such anomalies using only constant memory and computation.

Our experiment consists of two phases. In the first phase, we train Reverse GVFs off-policy. Our behavior policy $\mu$ is uniformly random with $\mu(a_1|s) = 0.5 \, \forall s$. The target policy is $\pi$ with $\pi(a_1|s) = 0.1 \, \forall s$. Given a transition $(s, a, r, s')$ following $\mu$, we update the parameters $\theta$, which is a look-up table in this experiment, to minimize $\rho(s,a)L(\theta)$. In this way, we approximate $\eta_\pi^s$ with $N = 20$ quantiles for all $s$. The MVE against training steps is reported in Figure 3a.

In the second phase, we use the learned $\eta_\pi^s$ from the first phase for anomaly detection when we actually deploy $\pi$. Namely, we let the microdrone follow $\pi$ for $2 \times 10^4$ steps and compute $\bar{G}_t$ on the fly. In the first $10^4$ steps, there is no anomaly. In the second $10^4$ steps, the aforementioned *reward anomaly* or *policy anomaly* happens every step. We aim to discover the anomaly from the information provided by $\bar{G}_t$ and $\eta_\pi^{S_t}$. Namely, we report the probability of anomaly as

$$\text{prob}_{\text{anomaly}}(\bar{G}_t) \doteq 1 - \eta_\pi^{S_t}([\bar{G}_t - \Delta, \bar{G}_t + \Delta]),$$

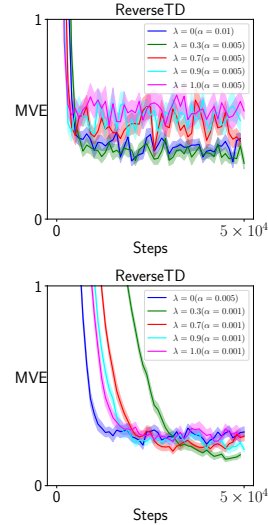

Figure 2: Left: the step size $\alpha$ is tuned to minimize the area under the curve, a proxy for learning speed. Right: the step size $\alpha$ is tuned to minimize the MVE at the end of training. All curves are averaged over 30 independent runs with shaded regions indicate standard errors.

where $\Delta$ is a hyperparameter. If a larger $\Delta$ is used, the reported probability of anomaly will in general be closer to 0, but then the algorithm becomes less sensitive to anomaly (e.g., if $\Delta$ is $\infty$, the output will always be 0). So $\Delta$ achieves a trade-off between reducing the false alarms (i.e., making the output as low as possible when no anomaly) and increasing sensitivity to the anomaly. This approach for computing the probability of anomaly is simple but intuitive. A more formal approach requires properly defined priors over $\bar{G}_t$ and the occurrence of anomalies to make use of Bayes' rule. However, those priors depend heavily on the application and complicate the presentation of the central idea to conduct anomaly detection with reverse GVF. We, therefore, use this simple approach in our paper. We believe detecting anomaly using only a single observation based on a known p.d.f. itself is an interesting statistical problem that is out of the scope of this paper. We use $\Delta = 1$ in our experiments. Moreover, we do not have access to $\eta_\pi^{S_t}$ but only $N$ estimated quantiles $\{q_i(S_t; \theta)\}_{i=1,\ldots,N}$. To compute $\text{prob}_{\text{anomaly}}(\bar{G}_t)$, we need to first find a distribution whose quantiles are $q_i(S_t; \theta)$. This operation is referred to as imputation in Rowland et al. (2018). Such a distribution is not unique. The commonly used imputation strategy for quantile-regression-based distributional RL is $\frac{1}{N} \sum_{i=1}^N \delta_{q_i(S_t;\theta)}$ (Dabney et al., 2017). This distribution, however, makes it difficult to compute $\text{prob}_{\text{anomaly}}(\bar{G}_t)$. Inspired by the fact that a Dirac can be regarded as the limit of a normal distribution with a decreasing standard derivation, we define our approximation for $\eta_\pi^{S_t}$ as $\hat{\eta}_\pi^{S_t} \doteq \frac{1}{N} \sum_{i=1}^N \mathcal{N}(q_i(S_t; \theta), \sigma^2)$, where $\sigma$ is a hyperparameter and we use $\sigma = 1$ in our experiments. Note $\hat{\eta}_\pi^{S_t}$ does not necessarily have the quantiles $q_i(S_t; \theta)$. We report $1 - \hat{\eta}_\pi^{S_t}([\bar{G}_t - \Delta, \bar{G}_t + \Delta])$ against time steps in Figure 3b. When the anomaly occurs after the first $10^4$ steps, the probability of anomaly reported by Reverse GVF becomes high.

*Non-linear Function approximation.* We now consider `Reacher` from OpenAI gym (Brockman et al., 2016) and use neural networks as a function approximator for $q_i(s; \theta)$. Our setup is the same

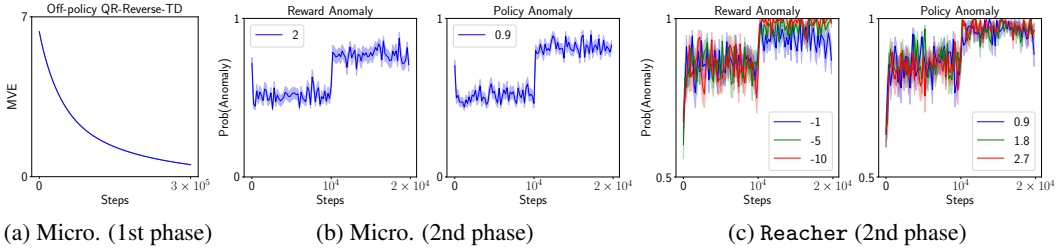

(a) Micro. (1st phase)　　　(b) Micro. (2nd phase)　　　(c) `Reacher` (2nd phase)

Figure 3: All curves are averaged over 30 independents runs with shaded regions indicate standard errors. (a) MVE against training steps in the first phase of the microdrone example. (b) Anomaly probability in the second phase of the microdrone example. (c) Anomaly probability in the second phase of `Reacher`, with three different reward anomalies and policy anomalies.

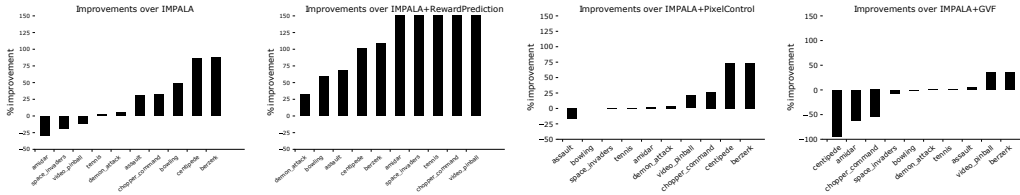

Figure 4: The performance improvement of IMPALA+ReverseGVF over plain IMPALA, IMPALA+RewardPrediction, IMPALA+PixelControl, and IMPALA+GVF. All agents are trained for $2 \times 10^8$ steps. The performance of an algorithm A, denoted as perf(A), is computed as the evaluation performance at the end of training. The improvement of $A_1$ over $A_2$ is computed as $\frac{\text{perf}(A_1) - \text{perf}(A_2)}{|\text{perf}(A_2)|}$. The results are averaged over 3 seeds.

as the tabular setting except that the tasks are different. For a state $s$, we define $\gamma(s) = 0$ if the distance between the end of the robot arm and the target is less than 0.02. Otherwise we always have $\gamma(s) = 1$. When the robot arm reaches a state $s$ with $\gamma(s) = 0$, the arm and the target are reinitialized randomly. We first train a deterministic policy $\mu_d$ with TD3 (Fujimoto et al., 2018) achieving an average episodic return of $-4$. In the first phase, we use a Gaussian behavior policy $\mu(s) \doteq \mathcal{N}(\mu_d(s), 0.5^2)$. The target policy is $\pi(s) \doteq \mathcal{N}(\mu_d(s), 0.1^2)$. In the second phase, we consider two kinds of anomaly. In the *policy anomaly*, we consider three settings where the policy $\pi(s)$ becomes $\mathcal{N}(\mu_d(s), 0.9^2), \mathcal{N}(\mu_d(s), 1.8^2)$, and $\mathcal{N}(\mu_d(s), 2.7^2)$ respectively. In the *reward anomaly*, we consider three settings where with probability 0.5 the reward $R_t$ becomes $R_t - 1$, $R_t - 5$, and $R_t - 10$ respectively. We report the estimated probability of anomaly in Figure 3c. When an anomaly happens after the first $10^4$ steps, the probability of anomaly reported by Reverse GVF becomes high. In Figure 3c, the probability of anomaly is higher than 0.8. This is mainly due to the larger variance of the observed reward (compared with the toy MDP used in Figure 3b), resulting from the large stochasticity of the policy being followed. When the variance of a random variable is large, the probability mass is not concentrated. Consequently, the information that a single observation can provide is less. So our anomaly detection has a higher chance for a false alarm.

We note that the goal of this work is not to achieve a new state-of-the-art in anomaly detection. Simple heuristics are enough to outperform our approach in the tested domains. Instead, we want to highlight the potential of Reverse-RL-based anomaly detection. An agent can obtain and maintain a huge amount of retrospective knowledge easily via off-policy Reverse RL with function approximation. Given the learned retrospective knowledge, anomaly detection can be simple and cheap. Our empirical study aims to provide a proof-of-concept of this new paradigm. There are indeed open questions in this new paradigm, e.g., the possible large variance of $\bar{G}_t$ and the threshold for anomaly alert, which we leave for future work.

**Representation Learning.** Veeriah et al. (2019) show that automatically discovered GVFs can be used as auxiliary tasks (Jaderberg et al., 2016) to improve representation learning, yielding a performance boost in the main task. Let $r$ and $\gamma$ be the reward function and the discount factor of the main task. Veeriah et al. (2019) propose two networks for solving the main task: a main task

and answer network, parameterized by $\theta$, and a question network, parameterized by $\phi$. The two networks do not share parameters. The question network takes as input states and outputs two scalars, representing a reward signal $\hat{r}$ and a discount factor $\hat{\gamma}$. The $\theta$-network has two heads with a shared backbone. The backbone represents the internal state representation of the agent. One head represents the policy $\pi$, as well as the value function $v_{\pi,r,\gamma}$, for the main task. The other head represents the answer to the *predictive question* specified by $\pi, \hat{r}, \hat{\gamma}$, i.e., this head represents the value function $v_{\pi,\hat{r},\hat{\gamma}}$. At time step $t$, $\theta$ is updated to minimize two losses $L_{\text{RL}}(\theta_t)$ and $L_{\text{GVF}}(\theta_t)$. Here $L_{\text{RL}}(\theta_t)$ is the usual RL loss for $\pi$ and $v_{\pi,r,\gamma}$, e.g., Veeriah et al. (2019) consider the loss used in IMPALA (Espeholt et al., 2018). $L_{\text{GVF}}(\theta_t)$ is the TD loss for training $v_{\pi,\hat{r},\hat{\gamma}}$ with $\hat{r}$ and $\hat{\gamma}$. Minimizing $L_{\text{RL}}(\theta_t)$ improves the policy $\pi$ directly, and Veeriah et al. (2019) show that minimizing $L_{\text{GVF}}(\theta_t)$, the loss of the auxiliary task, facilitates the learning of $\pi$ by improving representation learning. Every $K$ steps, the question network is updated to minimize $L_{\text{meta}}(\phi) \doteq \sum_{i=t-K}^{t} L_{\text{RL}}(\theta_i)$. In this way, the question network is trained to propose useful predictive questions for learning the main task.

We now show that automatically discovered Reverse GVFs can also be used as auxiliary tasks to improve the learning of the main task. We propose an IMPALA+ReverseGVF agent, which is the same as the IMPALA+GVF agent in Veeriah et al. (2019) except that we replace $L_{\text{GVF}}(\theta_t)$ with $L_{\text{ReverseGVF}}(\theta_t)$. Here $L_{\text{ReverseGVF}}(\theta_t)$ is the Reverse TD loss for training the reverse general value function $\bar{v}_{\pi,\hat{r},\hat{\gamma}}$ with $\hat{r}$ and $\hat{\gamma}$, and the $\bar{v}_{\pi,\hat{r},\hat{\gamma}}$-head replaces the $v_{\pi,\hat{r},\hat{\gamma}}$-head in Veeriah et al. (2019). We benchmark our IMPALA+ReverseGVF agent against a plain IMPALA agent, an IMPALA+RewardPrediction agent, an IMPALA+PixelControl agent, and an IMPALA+GVF agent in ten Atari games. [4] The IMPALA+RewardPrediction agent predicts the immediate reward of the main task of its current state-action pair as an auxiliary task (Jaderberg et al., 2016). The IMPALA+PixelControl agent maximizes the change in pixel intensity of different regions of the input image as an auxiliary task (Jaderberg et al., 2016).

The results in Figure 4 show that IMPALA+ReverseGVF yields a performance boost over plain IMPALA in 7 out of 10 tested games, and the improvement is larger than 25% in 5 games. IMPALA+ReverseGVF outperforms IMPALA+RewardPrediction in all 10 tested games, indicating reward prediction is not a good auxiliary task for an IMPALA agent in those ten games. IMPALA+ReverseGVF outperforms IMPALA+PixelControl in 8 out of 10 tested games, though the games are selected in favor of IMPALA+PixelControl. IMPALA+ReverseGVF also outperforms IMPALA+GVF, the state-of-the-art in discovering auxiliary tasks, in 3 games. Overall, our empirical study confirms that ReverseGVFs are useful inductive bias for composing auxiliary tasks, though not achieving a new state of the art.

## 5   Related Work

The reverse return $\bar{G}_t$ is inspired by the followon trace $F_t$ in Sutton et al. (2016), which is defined as $F_t \doteq i(S_t) + \gamma(S_t)\rho_{t-1}F_{t-1}$, where $i : \mathcal{S} \rightarrow [0, \infty)$ is a user-defined interest function specifying user's preference for different states. Sutton et al. (2016) use the followon trace to reweight value function update in Emphatic TD. Later on, Zhang et al. (2020d) propose to learn the conditional expectation $\lim_{t\to\infty} \mathbb{E}[F_t|S_t = s]$ with function approximation in off-policy actor-critic algorithms. This followon trace perspective is one origin of bootstrapping in the reverse direction, and the followon trace is used only for stabilizing off-policy learning. The second origin is related to learning the stationary distribution of a policy, which dates back to Wang et al. (2007, 2008) in dual dynamic programming for stable policy evaluation and policy improvement. Later on, Hallak and Mannor (2017); Gelada and Bellemare (2019) propose stochastic approximation algorithms (discounted) COP-TD to learn the density ratio, i.e. the ratio between the stationary distribution of the target policy and that of the behavior policy, to stabilize off-policy learning. Our Reverse TD differs from the discounted COP-TD in that (1) Reverse TD is on-policy and does not have importance sampling ratios, while discounted COP-TD is designed only for off-policy setting, as there is no density ratio in the on-policy setting. (2) Reverse TD uses $R_t$ in the update, while discounted COP-TD uses a carefully designed constant. The third origin is an application of RL in web page ranking (Yao and Schuurmans, 2013), where a different reverse Bellman equation is proposed to learn the authority score function. Although the idea of reverse bootstrapping is not new, we want to highlight that this paper is the first to apply this idea for representing retrospective knowledge and show its utility in

anomaly detection and representation learning. We are also the first to use distributional learning in reverse bootstrapping, providing a probabilistic perspective for anomaly detection.

Another approach for representing retrospective knowledge is to work directly with a reversed chain like Morimura et al. (2010). First, assume the initial distribution $\mu_0$ is the same as the stationary distribution $d_\pi$. We can then compute the posterior action distribution given the next state and the posterior state distribution given the action and the next state using Bayes' rule: $\Pr(a|s') = \frac{\sum_s d_\pi(s)\pi(a|s)p(s'|s,a)}{d_\pi(s')}, \Pr(s|s',a) = \frac{d_\pi(s)\pi(a|s)p(s'|s,a)}{d_\pi(s')}$. We can then define a new MDP with the same state space $\mathcal{S}$ and the same action space $\mathcal{A}$. But the new policy is the posterior distribution $\Pr(a|s')$ and the new transition kernel is the posterior distribution $\Pr(s|s',a)$. Intuitively, this new MDP flows in the reverse direction of the original MDP. Samples from the original MDP can also be interpreted as samples from the new MDP. Assuming we have a trajectory $\{S_0, A_0, S_1, A_1, \ldots, S_k\}$ from the original MDP following $\pi$, we can interpret the trajectory $\{S_k, A_{k-1}, \ldots, A_0, S_0\}$ as a trajectory from the new MDP, allowing us to work on the new MDP directly. For example, applying TD in the new MDP is equivalent to applying the Reverse TD in the original MDP. However, in the new MDP, we no longer have access to the policy, i.e., we cannot compute $\Pr(a|s')$ explicitly as it requires both $d_\pi$ and $p$, to which we do not have access. This is acceptable in the on-policy setting but renders the off-policy setting infeasible, as we do not know the target policy at all. We, therefore, argue that working on the reversed chain directly is only feasible for on-policy learning.

Designing effective auxiliary tasks to facilitate representation learning is an active research area. The notion of side prediction dates back to Sutton (1995); Littman and Sutton (2002); Sutton et al. (2011). Jaderberg et al. (2016) use reward prediction and pixel control as auxiliary tasks. Distributional RL methods (e.g., Bellemare et al. (2017); Dabney et al. (2017)) define auxiliary tasks implicitly by learning the full distribution of the return. Bellemare et al. (2019) use adversarial value functions as auxiliary tasks based on the value function geometry. Dabney et al. (2020) learn the value functions of past policies as auxiliary tasks based on the value improvement path. Srinivas et al. (2020) use contrastive learning as auxiliary tasks given its success in computer vision (He et al., 2020; Chen et al., 2020). Zhang et al. (2020b) show that by ignoring a $\gamma^t$ term in actor-critic algorithm implementations, practitioners implicitly implement an auxiliary task. All those auxiliary tasks are, however, handcrafted. By contrast, Veeriah et al. (2019) propose to discover auxiliary tasks automatically via meta gradients. Veeriah et al. (2019) define auxiliary tasks in the form of GVFs given the generality of GVFs in representing predictive knowledge. In this paper, we show the limit of GVFs in representing retrospective knowledge. Consequently, we propose to define auxiliary tasks in the form of Reverse GVFs. Our empirical study confirms that Reverse GVFs are also a promising inductive bias for meta-gradient-based auxiliary task discovery.

Anomaly detection has been widely studied in machine learning community (e.g., see Chandola et al. (2009, 2010); Chalapathy and Chawla (2019)). Using (Reverse) RL for anomaly detection, however, appears novel and this work provides a proof-of-concept for this new paradigm.

## 6   Conclusion

In this paper, we present Reverse GVFs for representing retrospective knowledge and formalize the Reverse RL framework. We demonstrate the utility of Reverse GVFs in both anomaly detection and representation learning. Investigating Reverse-GVF-based anomaly detection with real world data and applying Reverse GVFs in web page ranking are possible directions for future work.

## Broader Impact

Reverse-RL makes it possible to implement anomaly detection with little extra memory. This is particularly important for embedded systems with limited memory, e.g., satellites, spacecrafts, microdrones, and IoT devices. The saved memory can be used to improve other functionalities of those systems. Systems where memory is not a bottleneck, e.g., self-driving cars, benefit from Reverse-RL-based anomaly detection as well, as saving memory saves energy, making them more environment-friendly.

Reverse-RL provides a probabilistic perspective for anomaly detection. So misjudgment is possible. Users may have to make a decision considering other available information as well to reach a

certain confidence level. Like any other neural network application, combining neural network with Reverse-RL-based anomaly detection is also vulnerable to adversarial attacks. This means the users, e.g., companies or governments, should take extra care for such attacks when making a decision on whether there is an anomaly or not. Otherwise, they may suffer from property losses. Although Reverse-RL itself does not have any bias or unfairness, if the simulator used to train reverse GVFs is biased or unfair, the learned GVFs are likely to inherit those bias or unfairness. Although Reverse-RL itself does not raise any privacy issue, to make a better simulator for training, users may be tempted to exploit personal data. Like any artificial intelligence system, Reverse-RL-based anomaly detection has the potential to greatly improve human productivity. However, it may also reduce the need for human workers, resulting in job losses.

## Acknowledgments and Disclosure of Funding

SZ is generously funded by the Engineering and Physical Sciences Research Council (EPSRC). This project has received funding from the European Research Council under the European Union's Horizon 2020 research and innovation programme (grant agreement number 637713). The experiments were made possible by a generous equipment grant from NVIDIA.

## Footnotes

*Correspondence to `shangtong.zhang@cs.ox.ac.uk`

[2]For a full treatment of GVFs, one can use a transition-dependent reward function $r : \mathcal{S} \times \mathcal{S} \times \mathcal{A} \to \mathbb{R}$ and a transition-dependent discount function $\gamma : \mathcal{S} \times \mathcal{S} \times \mathcal{A} \to [0, 1]$ as suggested by White (2017). In this paper, we consider $r : \mathcal{S} \times \mathcal{A} \to \mathbb{R}$ and $\gamma : \mathcal{S} \to [0, 1]$ for the ease of presentation. All the results presented in this paper can be directly extended to transition-dependent reward and discount functions.

[3] Code available at `https://github.com/ShangtongZhang/DeepRL`

[4]Those ten Atari games are the ten where the IMPALA+PixelControl agent achieves the largest improvement over the plain IMPALA agent over all 57 Atari games (Veeriah et al., 2019).

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
