[Supplementary Material]

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

## A    Failure in Representing Retrospective Knowledge with GVFs

One may consider answering Question 2 with GVF via setting L4 to be the initial state and terminating an episode when the microdrone gets to L1. Then the value of L4 seems to be the answer to Question 2. To understand how this approach fails, let us consider transitions L4 - L3 - L4 - L1. It then becomes clear that we are unable to design a Markovian reward for the transition L3 - L4. This reward has to be non-Markovian to cancel all previously accumulated rewards. To make the reward Markovian, one may augment the state space with the battery level, which significantly increases the size of the state space. More importantly, this renders off-policy learning infeasible. The transition kernel on this augmented state space depends on the original reward function. So we cannot use off-policy learning to learn a GVF associated with a different reward function, as changing the reward function changes the transition kernel on the augmented state space. We can, of course, include the information about the new reward function into the augmented space. This, however, indicates the size of the state space grows exponentially with the number of reward functions we want to consider in off-policy learning. There is even a deeper defect. Let us consider the setting where we have two charging stations, say L2 and L4. Then if we want to use GVF directly as aforementioned assuming the aforementioned issues could somehow be solved, we need to set the initial state to L2 and L4 respectively. We then solve the two MDPs and compute $v(\texttt{L2})$ and $v(\texttt{L4})$ respectively. Finally, we may need to compute $d_\pi(\texttt{L2})v(\texttt{L2}) + d_\pi(\texttt{L4})v(\texttt{L4})$ as the answer, where $d_\pi$ is the stationary distribution of the original MDP, which is, unfortunately, unknown. To summarize, there may be some retrospective knowledge that GVF can represent if enough tweaks are applied. But in general, representing retrospective with GVF suffers from poor generality and poor scalability.

## B    Proofs

**Lemma 1.** *(Corollary 6.1 in page 150 of Bertsekas and Tsitsiklis (1989)) If $Y$ is a square nonnegative matrix and $\rho(Y) < 1$, then there exists some vector $w \succ 0$ such that $||Y||_\infty^w < 1$. Here $\succ$ is elementwise greater and $\rho(\cdot)$ is the spectral radius. For a vector $y$, its $w$-weighted maximum norm is $||y||_\infty^w \doteq \max_i |\frac{y_i}{w_i}|$. For a matrix $Y$, $||Y||_\infty^w \doteq \max_{y \neq 0} \frac{||Yy||_\infty^w}{||y||_\infty^w}$.*

### B.1    Proof of Theorem 1

*Proof.* Given the similarity between $\bar{G}_t$ and the followon trace $F_t$ as discussed in Section 5, the existence of $\lim_{t \to \infty} \mathbb{E}_{\pi,p,r}[\bar{G}_t|S_t = s]$ can be established in exactly the same way as Zhang et al. (2019) establish the existence of $\lim_{t \to \infty} \mathbb{E}[F_t|S_t = s]$ in their Lemma 1. We therefore omit this to avoid verbatim repetition. We have

$$\bar{v}_\pi(s) \doteq \lim_{t \to \infty} \mathbb{E}[\bar{G}_t|S_t = s]$$
$$= \lim_{t \to \infty} \mathbb{E}[R_t + \gamma(S_{t-1})\bar{G}_{t-1}|S_t = s]$$
$$= \lim_{t \to \infty} \sum_{\bar{s}, \bar{a}} \Pr(S_{t-1} = \bar{s}, A_{t-1} = \bar{a}|S_t = s)\mathbb{E}[R_t + \gamma(S_{t-1})\bar{G}_{t-1}|S_{t-1} = \bar{s}, A_{t-1} = \bar{a}]$$

(Law of total expectation)

$$= \sum_{\bar{s}, \bar{a}} \frac{d_\pi(\bar{s})\pi(\bar{a}|\bar{s})p(s|\bar{s}, \bar{a})}{d_\pi(s)}\Big(r(\bar{s}, \bar{a}) + \gamma(\bar{s})\bar{v}_\pi(\bar{s})\Big) \quad \text{(Bayes' rule)} \tag{4}$$

The matrix form of Eq (4) is exactly $\bar{v}_\pi = D_\pi^{-1}\tilde{P}_\pi^\top \tilde{D}_\pi r + D_\pi^{-1}P_\pi^\top \Gamma D_\pi \bar{v}_\pi$, solving which leads to $\bar{v}_\pi = D_\pi^{-1}(I - P_\pi^\top \Gamma)^{-1}\tilde{P}_\pi^\top \tilde{D}_\pi r$. Assumption 1 implies $\rho(P_\pi^\top \Gamma) < 1$. As $Y_1 Y_2$ and $Y_2 Y_1$ have the same eigenvalues (e.g., see Theorem 1.3.22 in Horn and Johnson (2012)), we have $\rho(D_\pi^{-1}P_\pi^\top \Gamma D_\pi) = \rho(P_\pi^\top \Gamma D_\pi D_\pi^{-1}) < 1$. Lemma 1 then implies $\bar{\mathcal{T}}$ is a contraction mapping w.r.t. some weighted maximum norm. $\square$

### B.2    Proof of Proposition 1

We first state a lemma about the convergence of the following iterates

$$w_{t+1} = w_t + \alpha_t(A(Y_t)w_t + b(Y_t)),$$

where $\{Y_t\}$ is a Markov chain evolving in $\mathcal{Y}$, $w_t \in \mathbb{R}^K$, $A : \mathcal{Y} \to \mathbb{R}^{K \times K}$, $b : \mathcal{Y} \to \mathbb{R}^K$.

**Assumption 2.** *(Assumption 4.5 in Bertsekas and Tsitsiklis (1996))*
*(a) The step sizes $\alpha_t$ are nonnegative, deterministic, and satisfy $\sum_t \alpha_t = \infty, \sum_t^\infty \alpha_t^2 < \infty$.*
*(b) The chain $\{Y_t\}$ has a stationary distribution $p_\mathcal{Y}$.*
*(c) The matrix $\bar{A} \doteq \mathbb{E}_{y \sim p_\mathcal{Y}}[A(y)]$ is negative definite.*
*(d) There is a constant $C_0$ such that $||A(y)|| \leq C_0$ and $||b(y)|| \leq C_0$.*
*(e) There exists scalars $0 < C_1, 0 < \rho < 1$ such that*

$$||\mathbb{E}[A(Y_t)] - \bar{A}|| \leq C_1 \rho^t, \quad ||\mathbb{E}[b(Y_t)] - \bar{b}|| \leq C_1 \rho^t,$$

*where $\bar{b} \doteq \mathbb{E}_{y \sim p_\mathcal{Y}}[b(y)]$.*

**Lemma 2.** *(Proposition 4.8 in Bertsekas and Tsitsiklis (1996))*
*Under Assumption 2, $\lim_{t \to \infty} w_t = -\bar{A}^{-1}\bar{b}$ with probability 1.*

We now prove Theorem 1 via verifying Assumption 2 thus invoking Lemma 2.

*Proof.* We first consider a deterministic reward setting, i.e., we assume $R_{t+1} = r(S_t, A_t)$. The Reverse TD update Eq (1) can be rearranged as

$$w_{t+1} = w_t + \alpha_t(A(Y_t)w_t + b(Y_t)),$$

where $Y_t \doteq (X_{t-1}, A_{t-1}, X_t), y = (s, a, s'), A(y) \doteq \gamma(s)x(s')x(s)^\top - x(s')x(s')^\top, b(y) \doteq r(s, a)x(s')$. It is easy to verify that $\bar{A} \doteq \mathbb{E}_{y \sim p_\mathcal{Y}}[A(y)] = X^\top(P_\pi^\top \Gamma - I)D_\pi X$ and $\bar{b} \doteq \mathbb{E}_{y \sim p_\mathcal{Y}}[b(y)] = \tilde{P}_\pi^\top \tilde{D}_\pi r$. Assumption 2(a) is satisfied automatically. Obviously $\{Y_t\}$ is ergodic and its stationary distribution is $p_\mathcal{Y}(y) \doteq d_\pi(s)\pi(a|s)p(s'|s, a)$. Assumption 2(b) is now satisfied.

We now verify Assumption 2(c). Our proof is inspired by the proof of Lemma 6.4 in Bertsekas and Tsitsiklis (1996). Let $z \in \mathbb{R}^{|\mathcal{S}|}/\{0\}$, we aim to show $z^\top \bar{A} z < 0$. As $X$ has linearly independent columns, it suffices to show $z^\top D_\pi(\Gamma P_\pi - I)z < 0$. We have

$$||\Gamma P_\pi z||_{D_\pi}^2 = \sum_s d_\pi(s)\gamma(s)^2 \Big(\sum_{s'} P_\pi(s, s')z(s')\Big)^2 \leq \sum_{s,s'} d_\pi(s)\gamma(s)^2 P_\pi(s, s')z(s')^2$$

$$\leq \sum_{s,s'} d_\pi(s)P_\pi(s, s')z(s')^2 = \sum_{s'} d_\pi(s')z(s')^2 = ||z||_{D_\pi}^2,$$

where the first inequality comes from Jensen's inequality, whose equality holds iff all components of $z$ are the same scalar (referred to as $z_c \neq 0$). When that happens, we have $||\Gamma P_\pi z||_{D_\pi}^2 = z_c^2 \sum_s d_\pi(s)\gamma(s)^2$. Note there exists at least one $s$ such that $\gamma(s) < 1$, otherwise $(\Gamma P_\pi - I)$ is singular, violating Assumption 1. So $||\Gamma P_\pi z||_{D_\pi}^2 < z_c^2 = ||z||_{D_\pi}^2$. To conclude, for any $z$, we always have $||\Gamma P_\pi z||_{D_\pi} < ||z||_{D_\pi}$, yielding

$$z^\top D_\pi \Gamma P_\pi z \leq ||z^\top D_\pi^{\frac{1}{2}}|| \, ||D_\pi^{\frac{1}{2}}\Gamma P_\pi z|| = ||z||_{D_\pi}||\Gamma P_\pi z||_{D_\pi} < ||z||_{D_\pi}^2 = z^\top D_\pi z,$$

which completes the proof.

Assumption 2(d) is straightforward as $\mathcal{Y}$ is finite. Assumption 2(e) is trivial in our setting as we do not have eligibility trace and can be obtained from standard arguments about the mixing time of MDP (e.g., Theorem 4.9 in Levin and Peres (2017)).

The extension from deterministic rewards to stochastic rewards is standard (e.g., see Section 2.2 in Borkar (2009)) thus omitted. □

### B.3 Proof of Proposition 2

*Proof.* Assumption 1 implies $\rho(P_\pi^\top \Gamma) < 1$. Then Lemma 1 implies that there exists a $w$ in $\mathbb{R}^{|\mathcal{S}|}$ such that $k_0 \doteq ||P_\pi^\top \Gamma||_\infty^w < 1$. Let $\ell_2$ be the Cramér distance in $\mathcal{P}(\mathbb{R})$ (see Definition 3 in Rowland et al.

(2018)), for any $\eta_1, \eta_2 \in (\mathcal{P}(\mathbb{R}))^{|\mathcal{S}|}$, we have

$$
\begin{aligned}
\ell_2^2\Big((\bar{\mathcal{T}}\eta_1)^s, (\bar{\mathcal{T}}\eta_2)^s\Big) &= \ell_2^2\Big(\int_{\mathbb{R}\times\mathcal{S}}(f_{r,\bar{s}}\#\eta_1^{\bar{s}})\mathrm{d}\,p(\bar{s},r|s), \int_{\mathbb{R}\times\mathcal{S}}(f_{r,\bar{s}}\#\eta_2^{\bar{s}})\mathrm{d}\,p(\bar{s},r|s)\Big) \\
&\leq \int_{\mathbb{R}\times\mathcal{S}}\ell_2^2\Big(f_{r,\bar{s}}\#\eta_1^{\bar{s}}, f_{r,\bar{s}}\#\eta_2^{\bar{s}}\Big)\mathrm{d}\,p(\bar{s},r|s) \\
&= \int_{\mathbb{R}\times\mathcal{S}}\gamma(\bar{s})\ell_2^2\Big(\eta_1^{\bar{s}},\eta_2^{\bar{s}}\Big)\mathrm{d}\,p(\bar{s},r|s) \\
&= \sum_{\bar{s}} p(\bar{s}|s)\gamma(\bar{s})\ell_2^2\Big(\eta_1^{\bar{s}},\eta_2^{\bar{s}}\Big), \quad\quad\quad\quad\quad (5)
\end{aligned}
$$

where the inequality comes from Jensen's inequality and the next equality comes from a property of $\ell_2$. We refer the reader to the proof of Proposition 2 in Rowland et al. (2018) for details. Let $\ell_2^{2,\eta_1,\eta_2}$ be a vector in $\mathbb{R}^{|\mathcal{S}|}$ with the $s$-th element $\ell_2^{2,\eta_1,\eta_2}(s) \doteq \ell_2^2(\eta_1^s, \eta_2^s)$, the RHS of Eq (5) is then $(P_\pi^\top \Gamma \ell_2^{2,\eta_1,\eta_2})(s)$. We have

$$
\begin{aligned}
\max_s \ell_2^2\Big((\bar{\mathcal{T}}\eta_1)^s, (\bar{\mathcal{T}}\eta_2)^s\Big)/w_s &\leq \max_s(P_\pi^\top\Gamma\ell_2^{2,\eta_1,\eta_2})(s)/w_s = ||P_\pi^\top\Gamma\ell_2^{2,\eta_1,\eta_2}||_\infty^w \\
&\leq k_0||\ell_2^{2,\eta_1,\eta_2}||_\infty^w = k_0\max_s\ell_2^2\Big(\eta_1^{\bar{s}},\eta_2^{\bar{s}}\Big)/w_s,
\end{aligned}
$$

indicating

$$
\max_s\ell_2\Big((\bar{\mathcal{T}}\eta_1)^s, (\bar{\mathcal{T}}\eta_2)^s\Big)/\sqrt{w_s} \leq \sqrt{k_0}\max_s\ell_2\Big(\eta_1^{\bar{s}},\eta_2^{\bar{s}}\Big)/\sqrt{w_s}. \quad\quad (6)
$$

With $d(\eta_1,\eta_2) \doteq \max_s\ell_2\Big(\eta_1^s,\eta_2^s\Big)/\sqrt{w_s}$, we have

$$
\begin{aligned}
d(\eta_1,\eta_2) + d(\eta_2,\eta_3) &= \max_s\ell_2(\eta_1^s,\eta_2^s)/\sqrt{w_s} + \max_s\ell_2(\eta_2^s,\eta_3^s)/\sqrt{w_s} \\
&\geq \max_s\Big(\ell_2(\eta_1^s,\eta_2^s) + \ell_2(\eta_2^s,\eta_3^s)\Big)/\sqrt{w_s} \\
&\geq \max_s\ell_2(\eta_1^s,\eta_3^s)/\sqrt{w_s} = d(\eta_1,\eta_3).
\end{aligned}
$$

In other words, $d$ satisfies the triangle inequality, indicating $d$ is indeed a valid metric. Eq (6) then implies that $\bar{\mathcal{T}}$ is a $\sqrt{k_0}$-contraction in $d$. Standard fixed point theories then imply that $\bar{\mathcal{T}}$ has a unique fixed point, which we refer to as $\eta_\pi$.

As $\mu_0 = d_\pi$, we have $p(\bar{s},r|s) = \Pr(S_{t-1} = \bar{s}, R_t = r|S_t = s)$ holds for all $t$. Then Eq (2) implies that $\eta_t = \bar{\mathcal{T}}\eta_{t-1}$, from which $\lim_{t\to\infty}d(\eta_t,\eta_\pi) = 0$ follows directly. $\qquad\square$

### B.4 Proof of Proposition 3

*Proof.* The proof is the same as the proof of Theorem 1 except that we define

$$
\begin{aligned}
A(y) &\doteq \tau(s)\rho(s,a)\big(\gamma(s)x(s')x(s)^\top - x(s')x(s')^\top\big), \\
b(y) &\doteq \tau(s)\rho(s,a)r(s,a)x(s').
\end{aligned}
$$

As we consider off-policy setting, the stationary distribution of $\{Y_t\}$ is then $p_{\mathcal{Y}}(y) \doteq d_\mu(s)\mu(a|s)p(s'|s,a)$. It is easy to verify that we still have $\bar{A} \doteq \mathbb{E}_{y\sim p_{\mathcal{Y}}}[A(y)] = X^\top(P_\pi^\top\Gamma - I)D_\pi X$ and $\bar{b} \doteq \mathbb{E}_{y\sim p_{\mathcal{Y}}}[b(y)] = \tilde{P}_\pi^\top\tilde{D}_\pi r$. The rest is thus the same. $\qquad\square$

## C Experiment Details

### C.1 Anomaly Detection

The TD3 agent used for generating $\mu_d$ is the same as Fujimoto et al. (2018), which is trained for $2\times10^4$ steps in Reacher to achieve an average episodic return of -4. We use two hidden-layer neural networks with ReLU (Nair and Hinton, 2010) activation function. Each hidden layer consists of 64 units. The output layer has 20 units, representing 20 quantiles. We use an Adam (Kingma and Ba, 2014) optimizer with an initial learning rate $5\times10^{-3}$. The size of the experience replay buffer is $10^4$ and the mini-batch size is 128. We update the target network every 100 steps. We conduct our experiments on an Nvidia DGX-1 with PyTorch, though no GPU is used.

## C.2 Representation Learning

As our IMPALA+ReverseGVF agent is simply replacing the canonical TD loss in the IMPALA+GVF agent with the Reverse TD loss, we refer the reader to Veeriah et al. (2019) for all the implementation details.