[Reviews · NeurIPS 2020]

Review 1

Summary and Contributions: This paper studies the RL problem with retrospective knowledge (i.e., the influence of past events on the present state). The authors propose Reverse GVFs to represent such knowledge. The authors extend some RL algorithms to the Reverse RL setting, including Reverse TD, Distributional Reverse TD, and Off-policy Reverse TD. The authors theoretically prove the convergence of these algorithms under linear function approximation. The paper empirically demonstrates the utility of Reverse GVFs in both anomaly detection and representation learning.

Strengths: 1) This paper focuses on an interesting and practical case of reinforcement learning. Clear examples are provided to demonstrate the difference between predictive knowledge (general RL) and retrospective knowledge (this work), how RL with retrospective knowledge can be used in real-world applications, and why general RL algorithms (GVFs) fail to represent such knowledge. 2) The proposed formulation is reasonable and easy to follow. The formulation is general so that multiple existing RL algorithms can be extended to the Reverse RL setting. Theoretical analysis is given to justify the convergence of Reverse RL algorithms (under linear function approximation). 3) This paper about reinforcement learning as well as its applications is relevant to the NeurIPS community.

Weaknesses: No obvious weakness is noticed.

Correctness: I have some concerns about the empirical evaluation: in Figure 3 (b,c), it’s clearly to observe that the the anomaly is detected after 10^4 steps, since the likelihood becomes high (close to 1). However, is it true that such likelihood should be as be as low as possible if there’s no anomaly? Before 10^4 steps (no anomaly), the likelihood is already higher than 0.5.

Clarity: The paper is well written generally. In the experiment evaluation, it would be clearer to show the meaning of legends in Fig. 3(b).

Relation to Prior Work: In terms of reinforcement learning, related work is clearly discussed. However, in terms of the applied domains, e.g., anomaly detection, it would be better to describe related work, or provide some discussions, relevant to RL.

Reproducibility: Yes

Additional Feedback:


Review 2

Summary and Contributions: The paper presents Reverse General Value Function to represents the influence of possible past events on the present observation (aka, retrospective knowledge). The proposed solution builds on top the Bellman operator (Yu et al., 2018) and is empirically tested both in anomaly detection and representation learning.

Strengths: - The representation learning experiments show that IMPALA+ReverseGVF is beneficial on some games. - Results suggest that ReverseGVF can help in the detection of synthetic generated anomalies. - I like the recurring example, it really helped me to follow the exposition.

Weaknesses: - The exposition of the Anomaly detection experiment can be improved. As far as I understand a simple anomaly is simulated after 10^4 steps. That event triggers an higher estimation for prob(anomaly) from the model. However the probability before that event is really high as well, higher than 0.5. It is not clear to me why that's the case. - Although the authors argue against a comparison with IMPALA+GVF, I think that would strengthen the paper. Moreover, I would have appreciated a deeper analysis/discussion on why for some games ReverseGVF hurts performance and for others improvements are observed. Are there any characteristics in the games that are negative for reverseGVF?

Correctness: some comments in the above section

Clarity: There are some parts that might benefit further clarification. For instance, I would appreciate a sentence that clarify if Assumption 1 is reasonable in a real-word scenario. Also, in Figure 4 I would specify the metric for which you see an improvement.

Relation to Prior Work: yes

Reproducibility: No

Additional Feedback:


Review 3

Summary and Contributions: This paper proposes a concept of retrospective knowledge which requires modeling influence of possible past events on the present, and studies how to represent retrospective knowledge using reinforcement learning methods. Reverse general value functions are proposed to represent retrospective knowledge. Experiments are conducted to validate proposed methods on anomaly detection and representation learning.

Strengths: The idea of retrospective knowledge is interesting and will be useful for other scenarios besides anomaly detection. The problem formation and theoretical analysis are in a way that I think reasonable and elegant. Multiple experiments are conducted to explore the benefit of using reverse GVF for representation learning. The reinforcement learning community may benefit from the proposed method to improve representation learning.

Weaknesses: The experiments conducted in this paper used only using synthetic data. It will be more convincing to use real-world data, e.g automatic driving vehicle status anomaly detection. The results of the experiment using non-linear function approximation in figure 3c are not clearly explained. It seems that even before 10^4 steps, the probability of anomaly is high (more than 0.8). Curves in that figure are not readable enough to show the impacts of different setups.

Correctness: The methodology is sound.

Clarity: Most of the paper is clear. Lacks of analysis for Non-linear Function approximation experiments.

Relation to Prior Work: This paper builds well on related work.

Reproducibility: Yes

Additional Feedback: Any other potential applicable scenarios besides anomaly detection?

[Author Response · NeurIPS 2020]

Thanks for the insightful feedback. Here we make some general clarification followed by individual responses.

**(a) Probability of anomaly:** In this paper, we report $1 - \eta_\pi^{S_t}([\bar{G}_t - \Delta, \bar{G}_t + \Delta])$ as the probability of anomaly, where
$\eta_\pi^{S_t}$ is the probability density function of $\bar{G}_t$ and $\Delta$ is a hyperparameter. If we use a large $\Delta$, the reported probability of
anomaly before $10^4$ steps will be close to 0, but then the algorithm becomes less sensitive to anomaly (e.g., if $\Delta$ is $\infty$,
the output will always be 0). So $\Delta$ achieves a trade-off between reducing the false alarms (i.e., making the output as
low as possible when no anomaly) and increasing sensitivity to the anomaly.

This is a simple but intuitive approach. A more formal approach requires properly defined priors over $\bar{G}_t$ and the
occurrence of anomalies to make use of Bayes' rule. However, those priors depend heavily on the application and
complicate the presentation of the central idea to conduct anomaly detection with reverse GVF. Therefore, we use the
simple approach in our paper. We believe detecting anomaly using only a single observation based on a known p.d.f.
itself is an interesting statistical problem that is out of the scope of this paper. Our contribution is to propose a (reverse)
RL approach to efficiently learn the p.d.f. We will clarify all the above in the next revision.

**(b) Figure 3c:** In Figure 3c, the probability of anomaly is higher than 0.8. This is mainly due to the larger variance
of the observed reward (compared with the toy MDP used in Figure 3b), resulting from the large stochasticity of the
policy being followed. When the variance of a random variable is large, the probability mass is not concentrated.
Consequently, the information that a single observation can provide is less. So our anomaly detection has a higher
chance for a false alarm. If the variance of $\eta_\pi^{S_t}$ is large (intuitively, the curve of the p.d.f. is likely to be flatter), then
$1 - \eta_\pi^{S_t}([\bar{G}_t - \Delta, \bar{G}_t + \Delta])$ will in general be large for all $\bar{G}_t$ due to the large randomness. We will clarify all the above
in the next revision.

**(R1) Legend:** In Figure 3b, "2" means the reward becomes $R_t + 2$ from $R_t$ after the anomaly occurs at $10^4$ step. "0.9"
means the probability of taking $a_1$ becomes 0.9 (from 0.1) after the anomaly occurs at $10^4$ step. These are explained in
Line 229 - 243 and we will clarify this further in the next revision. **Related work:** We will include more discussion
about anomaly detection from the non-RL community in the next revision.

**(R2) Baseline:** Thanks for the insightful suggestion; we will include a comparison with IMPALA+GVF in the
next revision. We will also move the comparison with IMPALA+PixelControl from the appendix to the main text.
**Performance:** It is not entirely clear why some games are negative for Reverse GVF. Our initial conjecture is that it
is related to the planning horizon. In the 10 tested games, Amidar seems to require the longest planning horizon. As
Reverse GVF plans in the reverse direction, the representation for Reverse GVF may be less useful if we require longer
planning horizon for the original problem. We will discuss this more in the next revision. **Assumption1:** As long as
the problem we consider in the real world has a recurring structure, the ergodic assumption can usually be fulfilled,
e.g., when we consider a bus commuting between two cities. As long as we set $\gamma$ to 0 for any state, the inversion exists.
E.g., if we are interested in the fuel of the bus, we could set $\gamma(\texttt{gas station}) = 0$. We will clarify this more in the
next revision. **Metric:** Let $A$ and $B$ be two algorithms; we use $R_A$ to denote the average undiscounted episodic return
of the evaluation episodes at the end of training of the algorithm $A$. The improvement of $A$ over $B$ is computed as
$\frac{R_A - R_B}{|R_B|}$. We will clarify this in the next revision.

**(R3) Real-world data & other scenarios:** Thanks for your insightful suggestion. This paper serves as the first work
to introduce the reverse RL framework and establish its theoretical foundations. We, therefore, focus on providing
motivating and easy-to-understand examples with synthetic data. Using real-world data may require extra engineering
tricks that complicate the presentation of the central idea, which we therefore leave for future work. Reverse GVF can
possibly be used for control as well if we learn $\bar{q}_\pi$ instead of $\bar{v}_\pi$, which we think deviates from the goal of this paper
and we leave for future work. **Figure 3c:** Please see **(a) & (b)**. Moreover, we will plot the curves for different setups
in Figure 3c separately in the next revision to improve readability. The overall message we want to convey is that our
anomaly detection method is robust across all the tested setups.

[Meta-Review · NeurIPS 2020]

This paper proposes Reverse Generalized Value Functions (RGVF) to model the influence of past events on the current state. In addition to the theoretical analysis of this novel concept, experiments on anomaly detection and representation learning illustrate its potential benefits. All reviewers appreciated the clear presentation of the core idea, which might have the potential to lead to further applications beyond those found in this submission. That being said, there were a few concerns regarding the significance of the empirical results, which I would like to amplify as I believe this is definitely a significant weakness of the paper: 1: For anomaly detection there is no comparison at all. I realize that "anomaly detection in RL" is not a popular research field, but couldn't one use standard anomaly detection algorithms (or even just straightforward heuristics) and apply them to some quantities collected by the agent? For instance for the drone "policy anomaly" example, just collecting the frequency of each action over N timesteps, and checking regularly if it changes, should be enough to detect that something is going wrong. For the "reward anomaly", similarly, it might be enough to just keep track of the accumulated reward every N timesteps. Or, you could monitor the error of your critic (which predicts how much reward you should get in the future). Also, as mentioned by several reviewers, the plots in Fig. 3 show that although the anomaly change points can easily been seen visually from the plots, it remains unclear how to translate the proposed method into a reliable anomaly detector (how do you set the threshold? how do you deal with the large variance of the anomaly detection score?) 2: For representation learning the comparisons are only vs IMPALA (the main baseline), IMPALA+RewardPrediction (it is good to check how this performs, but clearly it is very bad), and IMPALA+PixelControl (a relevant baseline, but based on a 4-year-old technique). And these experiments are only performed on 10 Atari games (while it is well known that there is a lot of variance across the typical 57 games of the full benchmark). There is no comparison to more recent representation learning techniques for RL, nor any mention of them in the "Related Work" section. The raw scores are not provided (in main text or Appendix), making it impossible to compare to other work. Finally, comparison to IMPALA+GVF should have definitely been included. At least, the authors do acknowledge that the goal of their work is not really to improve on existing techniques: "our empirical study is just a proof of concept to show the utility of Reverse GVFs in representation learning and does not aim for a new state of the art in designing auxiliary tasks" (this is a comment on the representation learning section, which I consider to also apply to the anomaly detection section). But the current experiments clearly aren’t convincing enough to show that the novel concept of RGVFs are actually useful in practice (either to do something better, or to enable new applications). None of the three reviewers participated in the post-rebuttal discussion. In light of this, given that they all considered the paper worth accepting in spite of its lack of convincing empirical results, I will follow their original recommendation. But I encourage the authors to reflect on the issues mentioned above, and consider adding more experimental results / references / discusses to address some of them in the camera ready version.